# Control over the emerging chirality in supramolecular gels and solutions by chiral microvortices in milliseconds

Jiashu Sun[1,2], Yike Li[1,2], Fusheng Yan[3], Chao Liu[1,2], Yutao Sang[2,4], Fei Tian[1], Qiang Feng[1,2], Pengfei Duan [2,3], Li Zhang[2,4], Xinghua Shi [2,3], Baoquan Ding[2,3] & Minghua Liu[2,3,4]

The origin of homochirality in life is a fundamental mystery. Symmetry breaking and subsequent amplification of chiral bias are regarded as one of the underlying mechanisms. However, the selection and control of initial chiral bias in a spontaneous mirror symmetry breaking process remains a great challenge. Here we show experimental evidences that laminar chiral microvortices generated within asymmetric microchambers can lead to a hydrodynamic selection of initial chiral bias of supramolecular systems composed of exclusively achiral molecules within milliseconds. The self-assembled nuclei with the chirality sign affected by the shear force of enantiomorphic microvortices are subsequently amplified into almost absolutely chirality-controlled supramolecular gels or nanotubes. In contrast, turbulent vortices in stirring cuvettes fail to select the chirality of supramolecular gels. This study reveals that a laminar chiral microflow can induce enantioselection far from equilibrium, and provides an insight on the origin of natural homochirality.

[1] CAS Key Laboratory of Standardization and Measurement for Nanotechnology, CAS Center for Excellence in Nanoscience, National Center for Nanoscience and Technology, No. 11 ZhongGuanCun BeiYiTiao, Beijing 100190, China. [2] University of Chinese Academy of Sciences, 100049 Beijing, China. [3] CAS Key Laboratory of Nanosystems and Hierarchical Fabrication, CAS Center for Excellence in Nanoscience, National Center for Nanoscience and Technology, No. 11 ZhongGuanCun BeiYiTiao, Beijing 100190, China. [4] CAS Key Laboratory of Colloid, Interface and Chemical Thermodynamics, Institute of Chemistry, Chinese Academy of Sciences, No. 2 ZhongGuanCun BeiYiJie, Beijing 100190, China. These authors contributed equally: Jiashu Sun, Yike Li., Fusheng Yan, Chao Liu. Correspondence and requests for materials should be addressed to J.S. (email: sunjs@nanoctr.cn) or to B.D. (email: dingbq@nanoctr.cn) or to M.L. (email: liumh@iccas.ac.cn)

Chirality, which can be observed from the subatomic levels to galactic scales, is a fundamental characteristic in nature[1]. At the supramolecular level, chirality plays important roles in biological activities such as DNA duplication, protein folding, and enzyme catalysis[2, 3]. Intriguingly, some achiral molecules can evolve in size, complexity, and function through self-assembly into chiral supramolecular systems by spontaneous mirror symmetry breaking[4–9]. These self-assembly processes of achiral molecules might be responsible for the origin and early evolution of life, making them subjects of immense interest over the past several decades[10–12].

Symmetry breaking of achiral molecules in the self-assembly process could be biased by purely physical fields, such as hydrodynamic flow by stirring[13–20], circularly polarized light irradiation[21–25], or the combination of hydrodynamic, magnetic, thermal, or light factors[9, 26–30]. For example, clockwise (CW) or counterclockwise (CCW) stirring of a solution containing exclusively protonated achiral porphyrin molecules in cuvettes leads to the formation of J-aggregates (dye aggregates with an absorption band shifted to a longer wavelength of increasing sharpness than that of the monomer[31]) of opposite handedness after removing the vortex motion. The phenomenon has been interpreted as the effect of the

chiral polarization in a mirror symmetry breaking scenario[13, 19], but also that macroscopic hydrodynamic forces could shift the equilibrium of a racemic mixture of J-aggregates, driving a redistribution of chiral aggregates favored by the stirring[32]. Moreover, the enantiospecific reversible effect on J-aggregates in dynamic stirring conditions is reversible, due to the temporary chiral ordering of nanoparticles in the flows[33, 34]. In contrast to the flow-driven chirality in solutions, stirring during gelation fails to affect the supramolecular chirality in gels, which has been determined stochastically by initial spontaneous symmetry breaking[6]. On the basis of these contradictory observations, it remains an enigma how an efficient hydrodynamic force could be created and how it controls over the supramolecular chirality during the self-assembly process.

On the prebiotic Earth, rock micropores in the size range from tens to hundreds of micrometers are ubiquitous in the vicinity of hydrothermal vents in the ocean[35]. The velocity of flow near the vents could reach several meters per second, and there is a large change to create high-speed microvortices (length scale from tens to hundreds of micrometers) up to $10^4$ rpm for flows over rock micropores[36]. These microvortices may provoke the aggregation of concentrated molecules under non-equilibrium conditions,

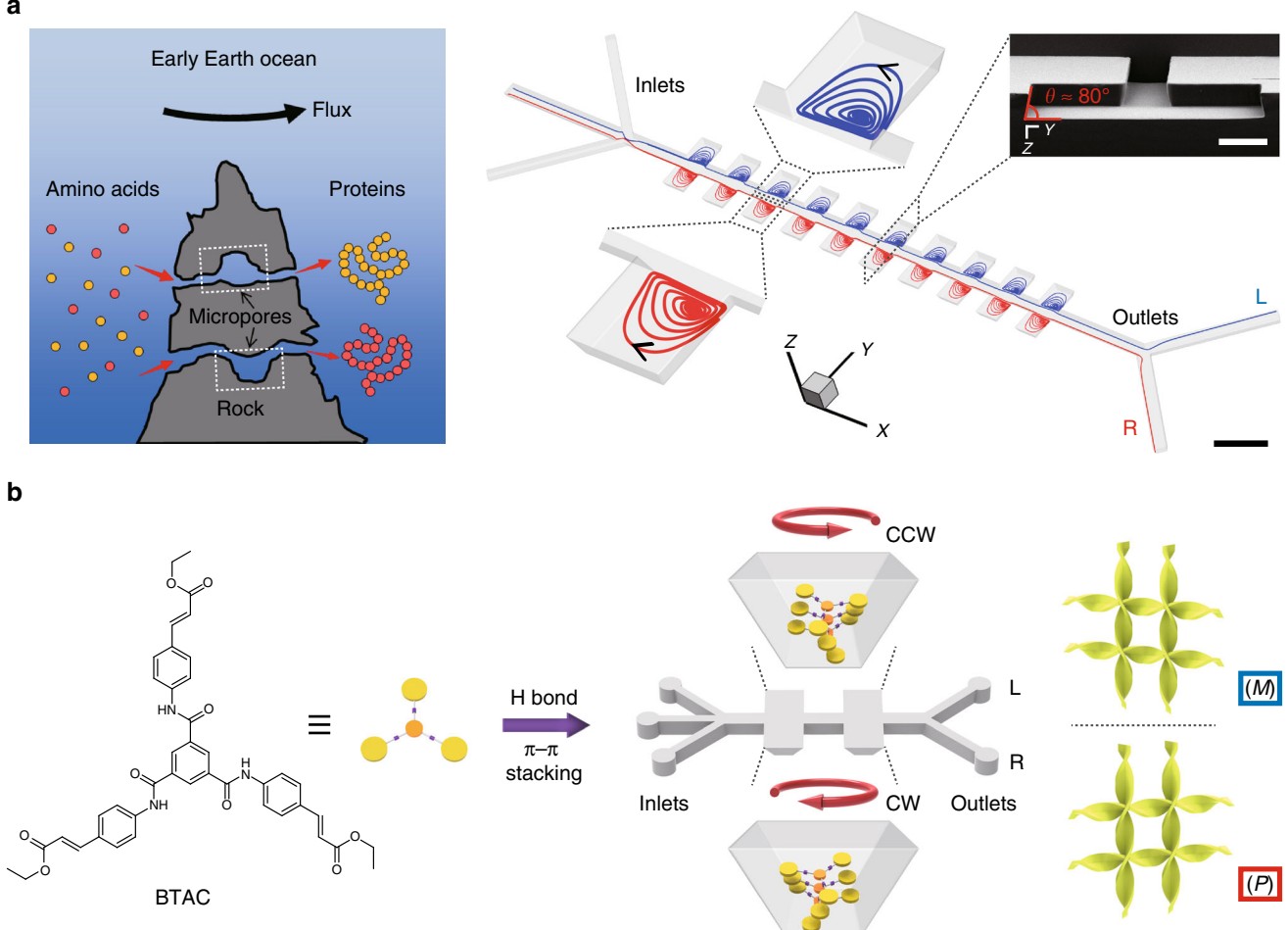

**Fig. 1** Microfluidic design for self-assembly of BTAC building blocks in microvortices. **a** Schematic hypothesis of the emergence of supramolecular chirality after molecules pass through the submarine rock micropores (left), and the imitated design of microfluidic device consisting of ten pairs of inclined microchambers (right) to generate counter-rotating microvortices. The blue (red) streamlines refer to CCW (CW) microvortices in the left (right) microchambers. L (R) indicates the left (right) outlet along the flow direction. Scale bar, 500 μm. The inset is SEM image of the cross-section of microchambers. Scale bar in the inset, 100 μm. **b** Effect of the shear forces of enantiomorphic microvortices (CW and CCW rotating directions) on the mirror symmetry breaking process of achiral BTAC molecules in DMF/$H_2O$ mixtures leading to the formation of BTAC gels. M (P) chirality refers to left-handed (right-handed) helical twists

leading to the emergence of supramolecular chirality, or even the appearance of living organisms[37–39].

Enlightened by natural rock micropores, we design a microfluidic device to induce counter-rotating laminar chiral microvortices (CW and CCW) inside opposing microchambers, to control over the supramolecular chirality of assemblies either in gels or solutions, exclusively from achiral molecules. The high magnitude of the shear gradient in laminar microvortices allows the alignment and twist of the supramolecular nuclei against the Brownian regime within milliseconds when mirror symmetry breaking occurs. These nuclei with a certain chiral bias correlated with the rotation sense of microvortices are subsequently amplified into supramolecular aggregates with nearly absolute chirality.

## Results

**Microvortex-induced self-assembly of BTAC.** The microfluidic device for generation of chiral microvortices consists of ten pairs of inclined microchambers with three inlets and two outlets (Fig. 1a and Supplementary Fig. 1). Two kinds of achiral

molecules, (tris(ethyl cinnamate) benzene-1,3,5-tricarboxamide, BTAC[6], and tetra-(4-sulfonatophenyl) porphyrin), TPPS$_4$[13], which have been reported to exhibit symmetry breaking during gel formation or aggregation, are selected. We first investigate the flow-induced self-assembly of BTAC (Fig. 1b). A solution of BTAC in $N,N$-dimethylformamide (DMF) and two individual DMF/H$_2$O mixtures (v/v, 5/2) are injected into the microfluidic device through the middle and two side inlets (Supplementary Figs. 2 and 3). Microvortex-induced symmetry breaking of CW- or CCW-rotated achiral BTAC molecules in DMF/H$_2$O mixtures leads to the formation of BTAC gels instantly. The gels are continuously collected from the right (R-) and left (L-) outlet. The optimization experiments are carried out to identify the critical conditions (BTAC concentration of 61 mg mL$^{-1}$ and temperature of 40 °C) in which the chirality of supramolecular systems is dependent on the rotation direction of microvortices (Supplementary Figs. 4 and 5). The broadened absorption peaks in the ultraviolet-visible (UV-vis) absorption spectra of BTAC gels are suggestive of $\pi$–$\pi$ stacking after self-assembly (Fig. 2a). The

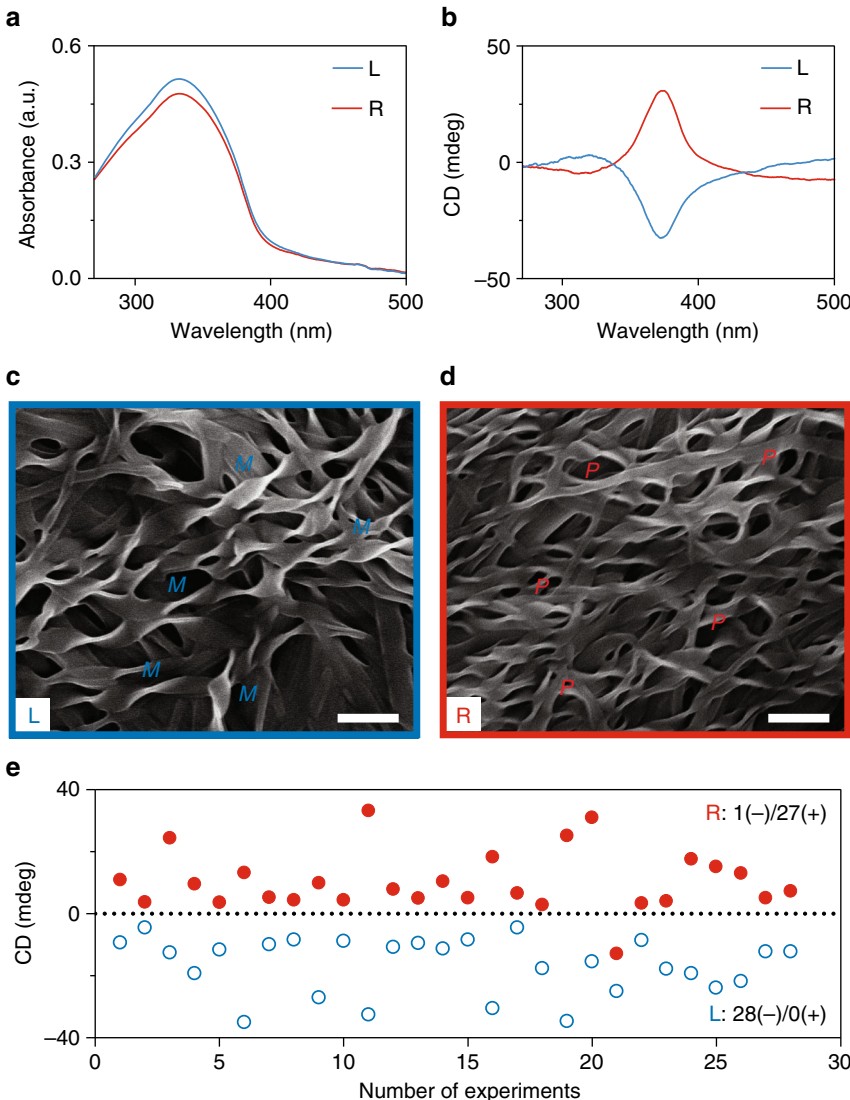

**Fig. 2** Microvortex selection of supramolecular chirality of BTAC gels. **a** UV-Vis spectra of BTAC gels. The blue (red) line refers to measurement from the gels collected from the L- (R-) outlet. **b** CD spectra indicate opposite chirality for BTAC gels collected from different outlets. **c**, **d** SEM characterizations show predominantly $M$ and $P$ twists for BTAC gels from the L- and R-outlet, respectively. Size of chiral twists is 295.2 ± 10.8 nm in width, and 869.3 ± 31.1 nm in length (mean ± s.e.m.; $n = 40$). Scale bars, 1 µm. **e** CD signals of 28 independent experiments indicate nearly absolute chirality control of BTAC gels by microvortices. The void blue (solid red) circles indicate CD signals of gels from the L- (R-) outlets

circular dichroism (CD) spectra show a negative cotton effect (the negative peak first appears as the wavelength decreases) for the L-outlet and a positive cotton effect (the positive peak first appears as the wavelength decreases) for the R-outlet (Fig. 2b), and no linear dichroism (LD) effects are detected in BTAC gels (Supplementary Fig. 6). Scanning electron microscopy (SEM) analysis indicates that the percentage of left-handed helical twists ($M$ chirality) is 80% ($n = 253$) in BTAC gels from the L-outlet, and that of right-handed helical twists ($P$ chirality) is 86% ($n = 271$) in gels from the R-outlet (Fig. 2c, d and Supplementary Fig. 7). The average size of chiral twists is 295.2 ± 10.8 nm in width, and 869.3 ± 31.1 nm in length from SEM images (mean ± s.e.m.; $n = 40$). The effect of microvortices on chiral selection in BTAC gels is confirmed by repeating the experiments 28 times using 28 microfluidic devices. The CD spectra of BTAC gels reveal 27 positive and 28 negative CD signals for the R- and L- outlets (Fig. 2e and Supplementary Fig. 8), verifying nearly absolute chirality control of BTAC gels by microvortices.

**Microvortex-induced self-assembly of TPPS₄.** The present microfluidic platform is extended to select the supramolecular chirality of the aggregates of TPPS₄ molecules (Fig. 3). Achiral molecules, TPPS₄ and 1-ethyl-3-methylimidazolium cations ($C_2mim^+$), are used as the building blocks and ionic stabilizer, respectively (Fig. 3a). Under acidic conditions (pH < 2), TPPS₄ molecules become positively charged via protonation and form J-aggregates by electrostatic and π–π stacking interactions. These interactions are promoted at a high ionic strength or in the presence of $C_2mim^+$[40]. With optimized microfluidic experimental parameters (Supplementary Fig. 9), TPPS₄ nuclei are formed within microvortices at the very beginning of the self-assembly process. These nuclei subsequently grow into TPPS₄ J-aggregates as observed in the UV-vis absorption spectral peaks at 491 nm

(Fig. 3b). A pair of bisignate CD signals of opposite signs is detected for TPPS₄ J-aggregates from the two outlets after growth (Fig. 3c). An almost absolute chirality control of TPPS₄ J-aggregates by microvortices is validated by the CD spectra of 28 independent experiments (Fig. 3d and Supplementary Fig. 10). The assembled TPPS₄ aggregates exhibit long hollow tubular nanostructures with an average diameter of 12.8 nm and a wall thickness of 2.4 nm (approximately the molecular length of TPPS₄), as determined by cryo-electron microscopy (cryo-EM), suggesting a helical arrangement of individual protonated TPPS₄ monomers (Fig. 3e, f). The assembly kinetics of TPPS₄ J-aggregates, as characterized by atomic force microscopy (AFM), reveals a rapid assembly of TPPS₄ monomers into a large number of primary nuclei (~260 nm long) within 5 min (Supplementary Fig. 11a), followed by subsequent growth into long, straight nanotubes (~1.7 μm long) after 2 h (Supplementary Fig. 11b). Together, our data indicate that microvortices can wield remarkable control over the supramolecular chirality far from equilibrium.

**Analysis of laminar chiral microvortices.** Characterizations of microvortices within inclined microchambers are performed by computational fluid dynamics (CFD) simulations. The mismatch of flow velocities $U$ between the main channel and the microchambers generates CCW (CW) laminar vortices in the left (right) microchamber (Fig. 4a). Simulated streamlines show that fluid within the left (right) microchamber could rotate upward or downward to generate a $P$ ($M$) or $M$ ($P$) microvortex (Supplementary Fig. 12). Interestingly, the majority (~84 %) of microvortices in the left (right) microchamber tend to spiral upward, showing a predominantly $P$ ($M$) chirality (Fig. 4b). The chiral microvortices also exhibit high rotation speeds reaching up to ~$10^4$ rpm, as estimated by $30u_j/\pi w$ (the average flow velocity $u_j$ is 1.9 m s⁻¹ at the main channel/microchamber junction, and the microchamber width $w$ is

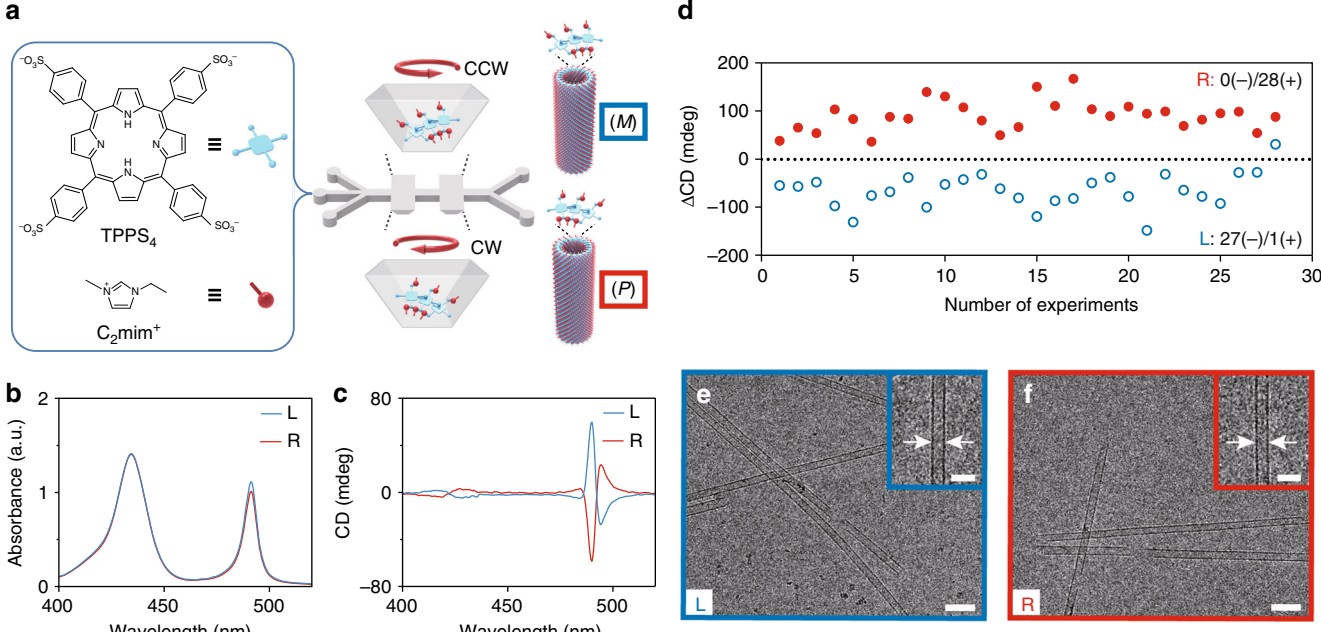

**Fig. 3** TPPS₄ J-aggregates with opposite chirality signs selected by microvortices. **a** Formation of chiral supramolecular nanotubes by the self-assembly of achiral TPPS₄ building blocks (blue) and the $C_2mim^+$ ionic stabilizer (red) within the microvortices. **b** UV-Vis spectra exhibit a monomer peak at 433 nm and a J-aggregate peak at 491 nm for TPPS₄ assemblies from the L- (blue line) and R-outlet (red line). **c** CD spectra indicate opposite chirality of the TPPS₄ assemblies from the L- (blue line) and R-outlet (red line). **d** CD signals of 28 independent experiments reveal nearly absolute chirality control over supramolecular TPPS₄ assemblies from the L- (void blue circles) and R-outlet (solid red circles). **e, f** Cryo-EM examinations of TPPS₄ aggregates from the L- and R-outlet show the hollow tubular nanostructures with an average diameter of 12.8 ± 0.1 nm (mean ± s.e.m.; $n = 40$). Scale bars, 50 nm. The insets are the enlarged cryo-EM images. Scale bars in the insets, 20 nm

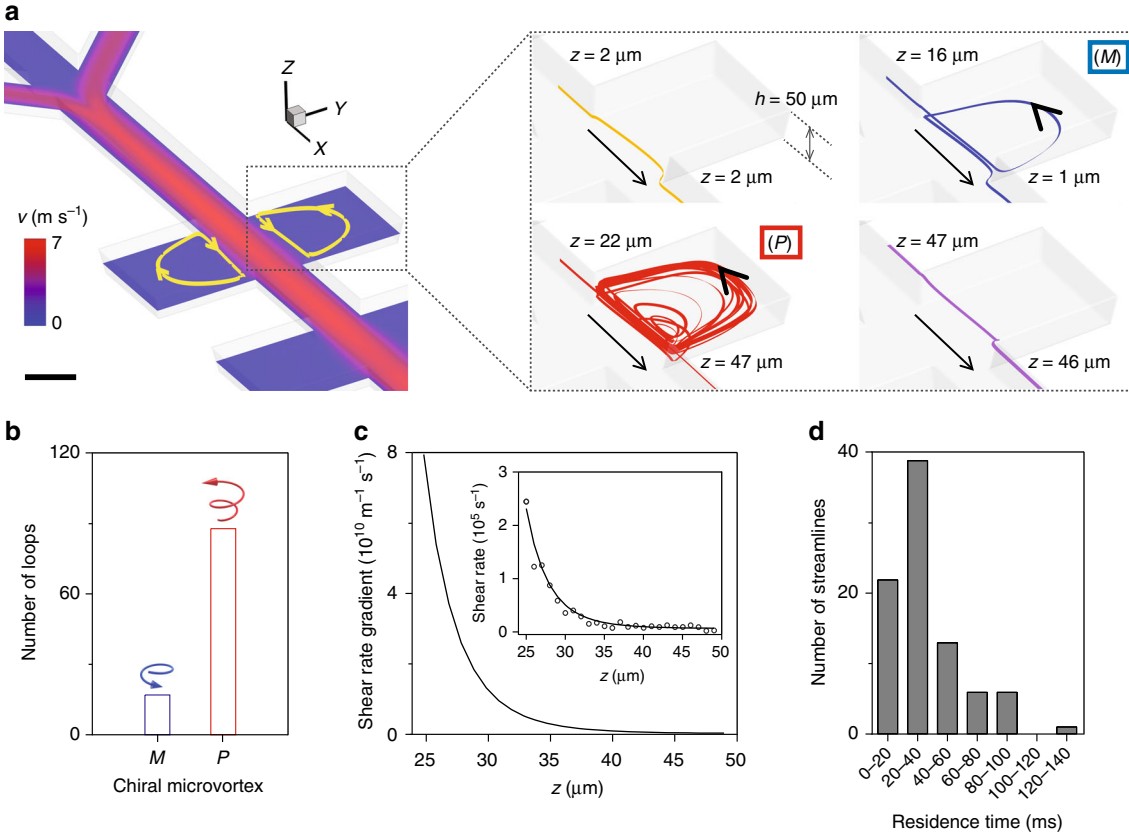

**Fig. 4** Analysis of laminar chiral microvortices. **a** CFD simulation shows that fluid can rotate upward (red curves) or downward (blue curves) to generate laminar $P$- or $M$-chiral microvortices in the left microchamber. The color bar indicates the magnitude of flow velocity. The channel height is 50 μm. Scale bar, 100 μm. **b** A majority (~84%) of microvortices rotate upward, suggesting a predominantly $P$ chirality (red bar) in the left microchamber. **c** Shear rate gradient from $P$-chiral microvortices in the left microchamber. Insert: shear rate (black line) obtained from simulation results (void black circles). **d** Residence time (gray bars) of nuclei within the left microchamber from simulation results

200 μm). The highest rotation speed of $4 \times 10^4$ rpm is also confirmed by high-speed microscopic observation of microvortices (Supplementary Fig. 13). Despite the high rotation speed, these microvortices remain laminar with the Reynolds number of ~470 of the main straight channel and that of 380 of the microchamber (Supplementary Methods and Supplementary Fig. 3).

**Laminar chiral microvortices for chiral selection.** These rapidly rotating laminar chiral microvortices are of significance in the selection of chirality at the initial stage of self-assembly. The assembly process is initiated when achiral BTAC (TPPS$_4$) building blocks are rotated and diffused into the DMF/H$_2$O (HCl/C$_2$mim$^+$) solutions driven by microvortices. The assembly time is estimated as 0.7 ms for BTAC and 0.5 ms for TPPS$_4$, implying a rapid formation of primary nuclei inside the microchamber (Supplementary Methods)[41]. These instantly formed nuclei are aligned in the laminar flows, and be able to detect the velocity gradients due to the increase in size. A shear force ($F_s$) originated from the shear rate gradient is applied to the nuclei, with the maximum of 1.6 pN for BTAC nuclei and 0.1 pN for TPPS$_4$ nuclei on the basis of the maximal shear rate gradient of $8 \times 10^{10}$ m$^{-1}$ s$^{-1}$ (Fig. 4c and Supplementary Methods). $F_s$ competes with intermolecular interactions ($F_i \approx 1$ pN)[42, 43], thus exerting a net torque that is strong enough to twist the nuclei and to give rise to a chiral bias within tens of ms (Fig. 4d). The twist on the bottom of nuclei is stronger than that on the crown, attributed to the decreased $F_s$ along the increased microchamber height, resulting in the opposite signs between the chiral bias of nuclei and the rotation direction of microvortices (Fig. 5a). We should note that

the TPPS$_4$ monomers may also self-assemble into the two-dimensional flexible sheets with intrinsically chirality, and roll into the short nuclei of monolayered nanotube structures by the hydrodynamic effect[44, 45]. These collected nuclei then serve as the templates for subsequent growth into assemblies with the pre-defined chirality (Supplementary Fig. 14).

**Turbulent vortices in stirring cuvettes.** In contrast to laminar chiral microvortices, the flow in stirring cuvettes cannot control over the chirality sign of BTAC supramolecular gels, as evidenced by 20 independent experiments showing 9 positive CD signals and 11 negative CD signals under the CW stirring (Fig. 5b, c). Notice that the vortices in stirring cuvettes at ~1000 rpm are turbulent with the Reynolds number of ~4400 and the maximum shear rate gradient of ~$10^6$ m$^{-1}$ s$^{-1}$ (Fig. 5b and Supplementary Fig. 15). Previous investigations indicate that laminar microvortices arising from the sudden expansion–contraction regions in microchannels could be used for size-based capture and concentration of rare cancer cells in blood samples[46, 47]. However, these microvortices are not chiral because of the symmetric shear rate gradients in the $z$ direction in vertical microchannels, and may not be capable of exerting enantioselective controls on supramolecular aggregates.

**Discussion**
The natural origin of homochirality is a long-standing question. It has been proposed that symmetry breaking at the molecular level induced by an external chiral influence may lead to a small excess of one enantiomer, which can be significantly amplified by autocatalytic reactions to display homochirality under prebiotic

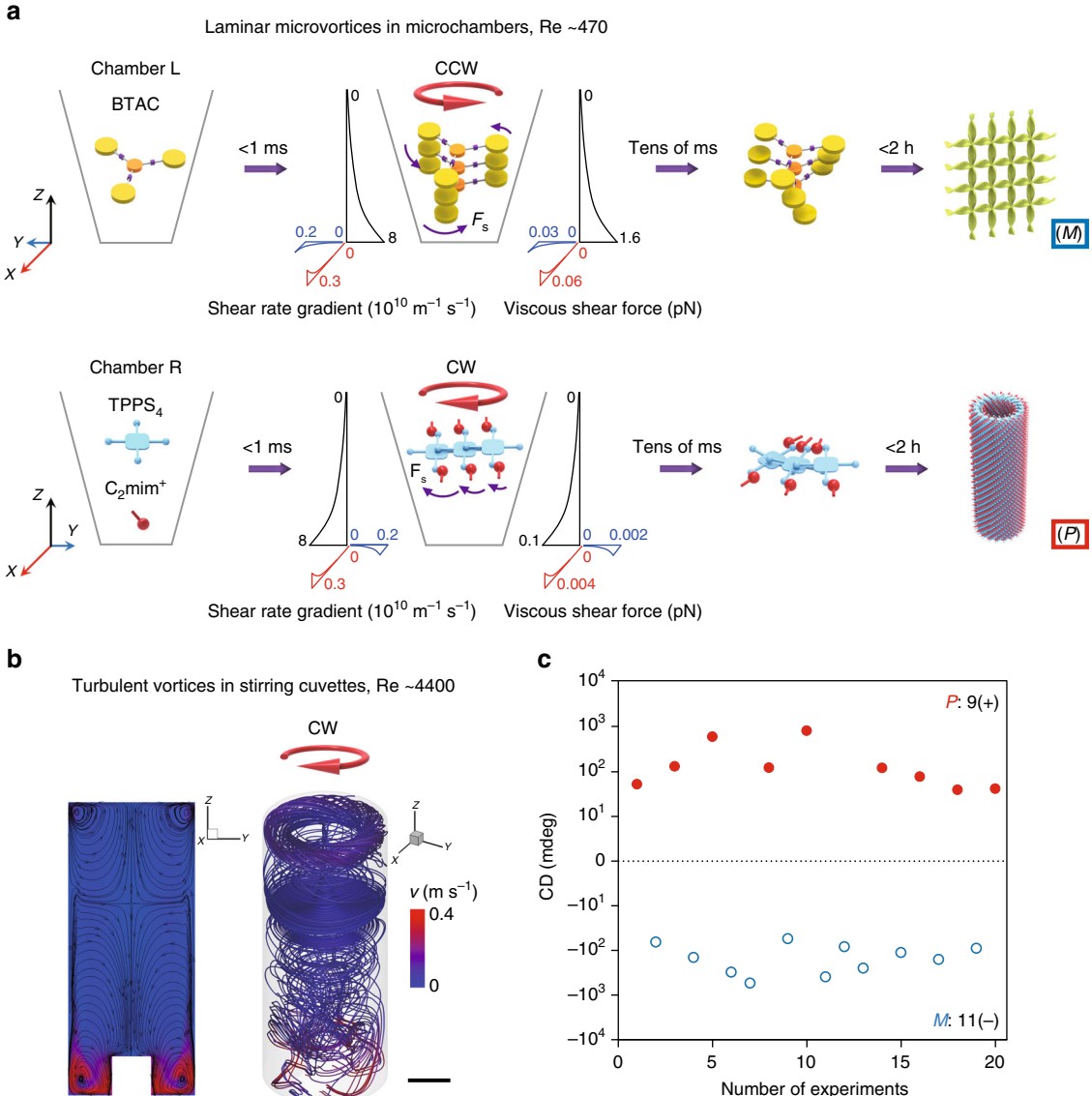

**Fig. 5** Laminar chiral microvortices versus turbulent vortices in stirring cuvettes. **a** Schematic mechanism of laminar chiral microvortex-selected chirality of BTAC and TPPS$_4$ nuclei. The rapidly formed nuclei (<1 ms) are aligned in the laminar flow, and twisted by a shear force ($F_s$) to give rise to an initial chiral bias dependent on the rotation sense of microvortices. These primary nuclei then serve as the templates for the subsequent growth into supramolecular assemblies with the predefined chirality. Red, blue, and black axes refer to the spatial distributions of shear rate gradient and viscous shear force along the x, y, and z axes. **b** CFD simulation of chaotic flows generated by CW stirring at ~1000 rpm in cuvettes. Left: 2D representation of the flow direction (black streamlines) and velocity (color). Right: 3D streamlines display a snapshot of chaotic flows. The color bar indicates the magnitude of flow velocity. Scale bar, 5 mm. **c** CW stirring in cuvettes shows 9 positive CD signals (P chirality, solid red circles) and 11 negative CD signals (M chirality, void blue circles) of BTAC gels

environments[48, 49]. Our study envisions a plausible scenario that prebiotic molecules could self-assemble into small nuclei/aggregates when passing through the natural asymmetric rock micropores in which laminar chiral microvortices exert a hydrodynamic torque on the nuclei. This chiral bias of nuclei appears almost instantly while being correlated with the rotation sense of microvortices. The subsequent propagation of this initial enantiomeric imbalance could result in the emergence of homochirality. This is analogous to a hypothesis that enantiomeric excesses might be induced by circularly polarized vacuum ultraviolet from space[50].

In this work, a versatile microfluidic platform has been presented to investigate the initial symmetry breaking and chiral selection in self-assembled systems of achiral molecules. The generated laminar chiral microvortices (P and M at 10$^4$ rpm) can exert a viscous shear force on instantly formed nuclei to compete with intermolecular forces, allowing for rapid selection of the initial chiral bias that is correlated with the rotation sense of microvortices within tens of millisecond. An almost absolute control (96%) over the chirality of BTAC gels or TPPS$_4$ nanotube solutions is realized after amplification. This study presents a distinct means for rapid and primary enantioselection far from equilibrium. We anticipate that our investigation may provide a hint as to the origins of natural homochirality, i.e., the critical role of oceanic vortices in protein folding and self-assembly in early Earth.

## Methods
**Materials**. 1,3,5-Benzenetricarbonyl trichloride, ethyl 4-aminocinnamate, trimethylamine, and tetrahydrofuran (THF) were purchased from Alfa Aesar (USA). BTAC monomers were synthesized by adding 1,3,5-benzenetricarbonyl trichloride

(0.2 M) in 10 mL THF solution to ethyl 4-aminocinnamate (0.4 M) and triethylamine (0.8 M) in 20 mL THF solution, followed by a 12-h stirring at room temperature. The filtrated product was concentrated by rotary evaporation and precipitated by adding 50 mL methanol. The solid precipitate was collected by filtration and washed consecutively with 20 mL water and 20 mL methanol. The product on the filter paper was dried to form powders. N,N-dimethylformamide (DMF) was purchased from Beijing Chemicals (China). 5,10,15,20-Tetraphenyl-21H, 23H-porphinetetrasulfonic acid, disulfuric acid, and tetrahydrate (TPPS$_4$·2H$_2$SO$_4$·4H$_2$O) were purchased from Dojindo Laboratories (Japan). 1-Ethyl-3-methylimidazolium tetrafluoroborate (C$_2$mimBF$_4$) was purchased from J&K (China). Hydrochloride acid (HCl, 36–38 wt %) was purchased from Beijing Chemicals (China). Polydimethylsiloxane (PDMS) was purchased from Dow Corning (USA). T3050 photoresists were provided by Baisiyou (China). Milli-Q water (18.2 MΩ cm at 25 °C) was used in all cases.

**Optimization of the flow condition**. A high flow rate ratio of the side inlet to the middle inlet facilitated a rapid bifurcation of the middle inlet solution to the sidewalls of the main channel[51, 52]. Using computational fluid dynamic (CFD) simulations and bright-field microscopy observations, the distributions of middle infusion were investigated under four different flow conditions: 10–1–10 mL h$^{-1}$, 20–1–20 mL h$^{-1}$, 30–1–30 mL h$^{-1}$, and 40–1–40 mL h$^{-1}$ for the side–middle–side inlets (Supplementary Figs. 2 and 3). Both the numerical and experimental observations suggested an optimal flow condition of 30–1–30 mL h$^{-1}$, under which the middle infusion was sufficiently bifurcated to enter into the microchambers. A further increase of the flow rate ratio to 40–1–40 mL h$^{-1}$ made the middle infusion to partially shift back to the channel centerline due to the over strong vortices at the inlet intersection. Therefore, we chose the flow condition of 30–1–30 mL h$^{-1}$ for the present study. These simulation and experimental results show that future work will be needed to properly characterize critical conditions for the flow-driven assembly in terms of appropriate dimensionless parameters when using a different microfluidic design.

**Self-assembly of BTAC gels in the inclined microchannels**. Using the inclined microchannels, 61 mg mL$^{-1}$ BTAC in DMF was infused through the middle inlet at 1 mL h$^{-1}$ while a mixture of DMF/H$_2$O (v/v, 5/2) was infused from the two side inlets at 30 mL h$^{-1}$ for each side. The flow rates were accurately controlled by syringe pumps (Harvard Apparatus, USA). Temperature was maintained at 40 °C by immersing the microfluidic devices in a water bath. Samples were collected 1 min after all the injections reached equilibrium. The samples separately collected from the two outlets were aged for 40 min at room temperature of 27 °C before spectra measurements.

**Procedure of TPPS$_4$ experiment**. TPPS$_4$·2H$_2$SO$_4$·4H$_2$O aqueous solution (20 μM) was injected into the microchannel through the two side inlets at a flow rate of 30 mL h$^{-1}$ for each side. The mixture solution (0.4 M C$_2$mimBF$_4$ + 1.5 M HCl) was injected through the middle inlet at a flow rate of 1 mL h$^{-1}$. The experimental observation and sample collection were performed 1 min after all the injections reached equilibrium. The samples separately collected from the two outlets were aged for 2 h before spectra measurements. All the TPPS$_4$ experiments were performed at room temperature of 27 °C.

**CFD simulation**. CFD simulations were performed to investigate the flow motion and species transportation at the experimental flow rates, which were obtained by solving the Navier–Stokes equations and species governing equation based on Fick's law:

$$\nabla \cdot \mathbf{u} = 0$$
$$\frac{\partial \mathbf{u}}{\partial t} + (\mathbf{u} \cdot \nabla)\mathbf{u} = -\frac{1}{\rho}\nabla p + \nu \nabla^2 \mathbf{u} \qquad (1)$$
$$\frac{\partial c_i}{\partial t} + \nabla \cdot (\mathbf{u}c_i) = \nabla \cdot (D_i \nabla c_i)$$

where $\mathbf{u}$ was the flow velocity vector, $\rho$ was the fluid density and set as $10^3$ kg m$^{-3}$, $p$ was the pressure, $\nu$ was the kinetic viscosity and set as $10^{-6}$ m$^2$ s$^{-1}$, $c_i$ was the species concentration, and $D_i$ was the mass diffusion coefficient of the species and set as $10^{-10}$ m$^2$ s$^{-1}$ [53]. All the governing equations were solved by commercial CFD software Fluent 6.4 (Ansys Inc., USA). The Navier–Stokes equations for the flow field were solved using the SIMPLE algorithm for pressure–velocity coupling. Both Navier–Stokes equations and species governing equation used second-order upwind scheme for spatial discretization. Hexahedral grids were generated using Gambit (Ansys Inc., USA) for inclined microchannels. Nonslip boundary conditions were implemented on the microchannel walls. Velocity inlet boundary conditions were applied for the three inlets to match the experimental flow rates and outflow boundary conditions were applied to the two outlets.

**Data availability**. The data that support the findings of this study are available from the corresponding authors on reasonable request.

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

## Acknowledgements

This work was supported financially by NSFC (21622503, 21475028, and 91427302) and Youth Innovation Promotion Association CAS (2016035). The authors thanked Dr. ZhenXi Guo from the Institute of Biophysics (CAS) for assistance with cryo-EM characterization.

## Author contributions

J.S., M.L., and B.D. conceived the idea. J.S., Y.L., F.Y., and Y.S. performed the experiments. C.L. performed the CFD simulation. J.S., Y.L., C.L., and F.T. created the figures. J. S., Y.L., C.L., Q.F., P.D., L.Z., X.S., B.D., and M.L. discussed the results. J.S., F.Y., C.L., and M.L. wrote the manuscript.

## Additional information

**Competing interests:** The authors declare no competing interests.

