## [Peer Review File · Nature Communications]

Reviewers' comments:

Reviewer #1 (Remarks to the Author):

Review on "Control over the emerging chirality

The manuscript is, from a methodological point of view, a sound innovative interdisciplinary work that shows definitive evidence on the effect of chiral mechanical forces (hydrodynamic forces) on the formation of chiral molecular structures from achiral building blocks. The work shows how the magnitude of the velocity gradients in laminar flows allows the alignment, or not, of the supramolecular particle in the flow against the Brownian regime. The high gradients obtained are what allow the particle alignment and the detection of the effect of hydrodynamic torques on the racemic bias when mirror symmetry breaking occurs.

The relatively few previous results obtained by shear hydrodynamic forces have been object of quite different explanations, even of skeptical considerations that have hindered the progress in the field. Some previous reports suggested already an explanation of the phenomenon that implies that microfluidics could lead to higher velocity gradients, i.e. higher hydrodynamic shear forces that could generalize the few obtained results in macroscopic vortices to a broader field of structures. How such chiral hydrodynamic forces could be created was quite unclear and the submitted work gives an answer to the problem; and the answer opens a broad prospect of future innovations.

The submitted work shows how the change of non-stochastic vortex directions in microfluidic lateral chambers created by viscous drag can be deterministic by designing inclined wall chambers, which create chiral velocity gradients. This is a novel approach that probably will be extended to many microfluidics applications involving the chirality and stereochemical aspects that define functionalities in molecular and supramolecular structures. The proposed scenario of the origin of biological homochirality in compartmentalization stages is, obviously, of interest, but the consequences of the presented results in supramolecular chemistry are, in my opinion, of high importance for many scientific fields. The work implies the use of chiral microfluidic hydrodynamic forces in the formation and/or separation of dissipative structures with a high control of the experimental conditions. In respect to the today available methods the work represent a qualitative change in the experimental stereochemical control of the formation of supramolecular dissipative structures (open thermodynamic systems). Notice that the use of consecutive microchambers along the microfluid direction is an approximation of an open system: a macroscopic open flow reactor cannot create enough intense gradients nor so perfect chiral flows (Fig. 5).

I strongly recommend the acceptance of the paper, but the authors must perform the following revisions before publication:

1) Those concerning the relationship of the presented results with previous work on the topic. This, because the presented results clarify previous controversies. In this respect, the first report in the field (ref.18) described already the phenomenon as the effect of a chiral polarization in a spontaneous mirror symmetry breaking (thermodynamic bifurcation; "chiral sign selection"). The low values of hydrodynamic forces do not justify a classical chiral induction. Therefore, the phrase in the Introduction (page 3 lines 18-19 and page 4 line one) can be only referred to ref. 34 but not to ref. 18. In contrast the lines 15-18 in page 3 should

contain the citation 18 in the context of the effect of a chiral polarization in a spontaneous mirror symmetry breaking process. For example changing the text to:

“For example, clockwise (CW) or counterclockwise (CCW) stirring of a solution containing exclusively protonated achiral porphyrin molecules leads to the formation of J-aggregates of opposite handedness. The phenomenon has been interpreted as the effect of the chiral polarization in a mirror symmetry breaking scenario^{18,23} but also that macroscopic hydrodynamic forces could shift the equilibrium of a racemic mixture of J-aggregates, driving a redistribution of chiral aggregates favored by the stirring^{18, (deleted), 34}.”

Notice too, that some authors interpreted the CD signal arising by the macroscopic chiral ordering in a macroscopic vortex (refs. 17, 21), but that only appears in dynamic stirring conditions. A final demonstration on this has been reported in *Nanoscale* 2015, 7, 20435. Obviously the objective of the paper is not a critical review of the topic, where the authors show important contributions, but it could be referred that the different interpretations of the phenomenon have been discussed in *The Chemical Record* 2017,17,1-13. This citation should substitute the older and less critical revision of ref. 23 (Chem Soc Rev).

In respect to the thermodynamic nature of the effect of the chiral polarization of stirring in a spontaneous mirror symmetry breaking process, this should be described in a more accurate way. For example in the caption of Fig. 1 the text (page 5, lines 7 and 8)

“b, Microvortex-induced symmetry breaking of CW- or CCW-rotated achiral...”

must be corrected, for example to

“b, Effect of the shear forces of enantiomorphic microvortexes (CW and CCW rotating directions) on the mirror symmetry breaking process of achiral BTAC molecules in DMF/H₂O mixtures leading to the formation of BTAC gels”

Further, such unclear descriptions of the mirror symmetry breaking also appear in the Abstract. The abstract should be rewritten. Now it does not? show the same quality level than that of the reported work.

2) The viscous drag generates the vortices, but for the chiral effect on the self-assembly the cause “upon” the particle is that of the shear forces. Laminar flows are necessary for the alignment of the growing particles, and the particle shape and length must be able to detect the velocity gradients. Viscous drag is the way how the device creates the vortexes but not the cause of the chiral effect upon the aggregate particles, which is the particle alignment and the existence of a strong enough torque. This is not clear in some parts of the text and should be revised.

The fig. 5 only tries to show the shear gradient in the Z and X direction of the microflow chamber, but for a chiral effect it must also exist a gradient in the Y direction. We surely agree with the authors that the gradients originating the torque on the particle aligned in the flow statistically follow the vectorial description of

I don't think that the present text gives a clear insight on this for a reader not working directly in the field.

3) Some references of previous reports describing similar devices with lateral microchambers for the generation of vortices, should be cited, but pointing their differences in respect to chamber device reported here.

4) In respect which the structural changes that may occur at the bifurcation point by effect of the hydrodynamic torques, I would like to comment some points, independently if they are taken or not into account in the revised text. The primary structure of the sulfonatoporphyrin J aggregates is that of a sheet (J. Mater. Sci. C 2013, 1, 33377) where the order porphyrin building blocks is of conformational racemic compositions, but the porphyrins show two twists, one inclination in the plane and other out of the sheet plane of the sheet. This converts the supramolecular structure in intrinsically chiral. In my opinion the hydrodynamic effect should occur at this stage of the particle growth. These sheets then close to the hollow nanotubes or other type of particles (Chem Eur J 2016, 22, 9749). The first symmetry breaking transfers to other stereomorphous aspects of the particles (for example nanotubes).

The work makes already some comments on the fact of such hierarchical chirality transfer. I suppose that the authors could emphasize this in some parts of the text.

4) Less important: The groups of accumulated cites (for example in page 3 13-15, 16-23, 24-28, etc) should appear in the References section in chronological order.

Josep M. Ribó (25/02/2018)

Reviewer #2 (Remarks to the Author):

The possible origins of prebiotic homochirality is a topic of great current interest. Stirring of solutions of achiral porphyrins has previously been shown to induce chiral supramolecular systems of J-aggregates with opposite handedness for clockwise or counterclockwise stirring. Although stirring of gels also generates supramolecular chirality, the handedness is determined stochastically rather than by the sense of stirring. The present paper extends this work to microvortices generated in asymmetric microchambers via a clever microfluidic device that produces supramolecular aggregates which, remarkably, show right- or left-handed chirality depending on the rotation sense for both solutions and gels. The experimental SEM data showing M and P twists of resulting BTAC gels are convincing, as are the associated mirror-image CD spectra which are reproducible over 27 out of 28 experiments. Likewise for the CD spectra of the generated J-aggregates, being reproducible over 28 out of 28 experiments. However, I do not have the expertise to assess the computational fluid dynamics simulations. A plausible scenario is suggested in which natural rock micropores might support this novel far-from-equilibrium enantioselection process, thereby providing possible new insight into the natural origin of homochirality. This well-written paper should be of interest to a wide audience.

Reviewer #3 (Remarks to the Author):

This paper offers interesting insights and experiments demonstrating the controlled formation of chiral aggregates in flows in microfluidic devices there the sense of the chirality is set by the sign of the vorticity in the microchambers. Results are shown for two different molecules, as given in Figure 2 for one molecule and Figure 3 for a second molecule. I do not know the background of this topic to know whether all of the results are original nor do I know whether it is significant that the results shown here are with gels whereas apparently some of flow drive chirality in solutions (in contrast to gels) seems to have been demonstrated (see item I below). The work is thought-provoking and I can see it being of interest to readers. There are four ideas that need clarification:

I) I sensed that the ideas likely are new but then I am a little confused by some of the references and descriptions, such as are not the experiments shown in this paper similar to the references made in the statement (p. 3) "For example, clockwise (CW) or counterclockwise (CCW) stirring of a solution containing exclusively protonated achiral porphyrin molecules leads to the formation of J-aggregates of opposite handedness." Hence, isn't this essentially what the authors say they accomplish when they write (p. 4): "we designed a microfluidic device to induce microscopic chiral factors, counter-rotating chiral microvortices (CW and CCW) inside opposing microchambers,

to control over the supramolecular chirality of assemblies either in gels or solutions, exclusively from achiral molecules (Fig. 1a and Supplementary Fig. 1)." So, I think it needs to be more clear whether and how the flow-driven chirality, so nicely demonstrated in this paper, is new?

II) Second, in the conclusions the authors write "We anticipate that our investigation may provide a hint as to the origins of natural homochirality, i.e. the critical role of oceanic vortices in protein folding and self-assembly in early Earth." But isn't the biological chirality observed on Earth largely one-signed? How do the authors think a flow feature, which would produce both left- and right-handed flows (signed vortices) could be responsible for selecting sign? Much as I find the demonstration of flow-driven chirality interesting, I think that it is unfair to claim relevance to biology without some plausible remarks about selection of "sign".

III) Third, maybe I misunderstood the SI but in supplementary figure 4 we read that a chiral-like signal is only obtained at 40C while in supplementary figure 5 a chiral-like signal is only obtained at one concentration. If so, this point is not clearly made in the main text. It also highlights that the "critical conditions" remain to be identified, which I can appreciate is a much longer term project but is something the authors can indicate to the reader.

IV) In my view the authors misuse the terms "microscopic" and "macroscopic" when discussing the features of the flow and their experiments. In fluid mechanics, as I think the authors realize given some of their discussion, a flow is characterized by the Reynolds number. Even "microscopic" systems, by which the authors may refer to systems that have dimensions 100s of microns can be turbulent if the flow is fast enough just as large-scale "macroscopic" flows can be laminar if the flow is slow enough (e.g. flows of glaciers). This microscopic/macroscopic language is often misused in the literature but here it is particularly unwise since (as supported by figure 5 in contrast to figures 2 and 3) the distinction in the authors results appears to be between laminar and turbulent flows (rather than between microscopic and macroscopic). It is also for this reason that the authors have to clarify item I above to demonstrate that their work is really new and not simply a "small scale" version of flow-driven chirality that already may have been demonstrated.

I believe this nice paper has a potentially original message to share but I think points I-IV above need clarification so that the results, previous work, and potential claims to the origin of chirality in biological systems are put into proper perspective.

Additional remarks:

1) p. 3: What are "J-aggregates"?

2) p. 4: "Microscopic chiral influences, such as chiral microflow, may impact on the selection of supramolecular chirality on a similar scale, and yet evidence is relatively scarce." - what does this mean? Has evidence been published? Since "micro" is not defined, this sounds similar to me to the previous statements in the introduction (p. 3, though "macro" is not defined): "Indeed, purely

physical fields, such as hydrodynamic flow by stirring¹⁶⁻²³, circularly polarized light irradiation²⁴⁻²⁸, or the combination of hydrodynamic, magnetic, thermal, or light factors^{9,29-33} can cause enantioselection in racemic mixtures by inducing an enantiomeric excess."

3) In fact, the remark about flows producing chiral structures from achiral starting features appears identical to me to one demonstration in this paper (p. 4): "The chiral microvortices enabled a primary selection of the chiral bias dependent on rotation sense".

4) p. 4: What are "M" and "P" chirality? This should be defined in the text (instead of the caption to figure 1).

5) "which have been reported to exhibit symmetry breaking during gel formation or aggregation" - what are the references? I guess they mean reference 35, which is indicated earlier in the sentence, but this is rather disingenuous for an article that uses a microfluidic flow to show (the title of the 2016 article) "Milliseconds make the difference in the far from equilibrium assembly of supramolecular chiral nanostructures".

6) P. 6: what is a "negative Cotton effect"?

7) p. 6: "Predominant amounts" - a quantitative approximation should be indicated.

8) Figure 2 is a very nice demonstration that the local flow controls the local chirality of the aggregates.

9) p. 12-13: The microfluidic chamber width is given but what is the height? If possible, give the dimensions in figure 4. As far as I can tell from the SI, the height of the channels was 50 microns.

10) p. 13: The statement about turbulence with a Reynolds number of 2000 applies to flow in a straight pipe and is not the right number of turbulence in flows with curved streamlines, which likely transitions to turbulence at a lower Reynolds number.

11) The shear rate gradient in figure 4c?

12) p. 12: what does "fluid rotations travelled upward" mean?

13) "new paradigm of chiral selection of supramolecular assemblies at the initial stage of self-assembly." - I agree this is shown but it is consistent

14) The assembly time is estimated as 0.7 ms for BTAC and 0.5 ms for TPPS4 - how is this estimated? How can the simulation of flow say anything about "assembly"? In the SI (p. 6) we read that the assembly time is calculated as $L^2/(6D)$ where L is the size of the primary nuclei but, while I agree this is a diffusion time, why should this necessarily be the assembly time? I

15) The authors write that the aggregates experience a "drag force" - this is not true in general for neutrally buoyant objects in flow. In the absence of a body force such as gravity these objects are force free. In the SI gravity is indicated but isn't this force much smaller than the other forces included in the simulation? Perhaps the point is that there are intermolecular forces with other molecules and then there can be a balancing fluid drag force. So the text has to be clarified.

16) Also, in the SI a drag force is calculated as $A \eta L \Gamma/2$ where Γ is the shear rate gradient - why is this quantity a drag force? I agree that the quantity has units of force but what fluid mechanical result indicates that this quantity is a drag force and is there a reference for this result or is it just an order of magnitude idea?

17) The twist on the bottom of nuclei was stronger than that on the crown, attributed to the decreased F_d along the increased microchamber height, resulting in the opposite signs between the chiral bias of nuclei and the rotation direction of microvortices (Fig. 5a).

18) In the concluding paragraph the authors return to remark about a possible role of rock micropores and refer to their experiments with 10^4 rpm. Is this realistic for flows over rock micropores?

19) 30 mL/hr; according to p. S1 of the SI the height of the channels was 50 microns and from figure 5 the widths of the microchamber and the main channel appear to be about 100 microns. The typical maximum speed in the main channel is then about $\langle u \rangle = 30 \text{ cm}^3/3600 \text{ sec}/(50 \times 10^{-4} \times 100 \times 10^{-4}) \text{ cm}^2 = (1/120)1/(2.5 \times 10^{-5}) = 10^5/300 = 3 \text{ m/s}$.

Based on this estimate, the experiments would have a typical $Re = 3 \text{ m/s} \times 25 \times 10^{-4} \text{ m} / 10^{-6} \text{ m}^2/\text{s} = 7500$, which is probably somewhat of an overestimate

The maximum gradient expected is then $3 \text{ m/s} / (25 \times 10^{-6} \text{ m}) = 10^5 \text{ sec}^{-1}$. These estimates seem comparable to the numerical results as far as I can tell.

20) The flow simulations seem fine. The specific values given show the idea and indicate that future work will be needed to properly characterize critical conditions for the flow-driven assembly in terms of appropriate dimensionless parameters.

Response to Reviewer #1

Comment: The manuscript is, from a methodological point of view, a sound innovative interdisciplinary work that shows definitive evidence on the effect of chiral mechanical forces (hydrodynamic forces) on the formation of chiral molecular structures from achiral building blocks. The work shows how the magnitude of the velocity gradients in laminar flows allows the alignment, or not, of the supramolecular particle in the flow against the Brownian regime. The high gradients obtained are what allow the particle alignment and the detection of the effect of hydrodynamic torques on the racemic bias when mirror symmetry breaking occurs.

The relatively few previous results obtained by shear hydrodynamic forces have been object of quite different explanations, even of skeptical considerations that have hindered the progress in the field. Some previous reports suggested already an explanation of the phenomenon that implies that microfluidics could lead to higher velocity gradients, i.e. higher hydrodynamic shear forces that could generalize the few obtained results in macroscopic vortices to a broader field of structures. How such chiral hydrodynamic forces could be created was quite unclear and the submitted work gives an answer to the problem; and the answer opens a broad prospect of future innovations.

The submitted work shows how the change of non-stochastic vortex directions in microfluidic lateral chambers created by viscous drag can be deterministic by designing inclined wall chambers, which create chiral velocity gradients. This is a novel approach that probably will be extended to many microfluidics applications involving the chirality and stereochemical aspects that define functionalities in molecular and supramolecular structures. The proposed scenario of the origin of biological homochirality in compartmentalization stages is, obviously, of interest, but the consequences of the presented results in supramolecular chemistry are, in my opinion, of high importance for many scientific fields. The work implies the use of chiral microfluidic hydrodynamic forces in the formation and/or separation of dissipative structures with a high control of the experimental conditions. In respect to the today available methods the work represent a qualitative change in the experimental stereochemical control of the formation of supramolecular dissipative structures (open thermodynamic systems). Notice that the use of consecutive microchambers along the microfluid direction is an approximation of an open system: a macroscopic open flow reactor cannot create enough intense gradients nor so perfect chiral flows (Fig. 5).

I strongly recommend the acceptance of the paper, but the authors must perform the following revisions before publication.

Response: We thank the reviewer for the positive evaluation of our research work. The following comments have been thoroughly addressed upon the reviewer's request.

Comment 1-1: Those concerning the relationship of the presented results with previous work on the topic. This, because the presented results clarify previous controversies. In this respect, the first report in the field (ref.18) described already the phenomenon as the effect of a chiral polarization in a spontaneous mirror symmetry breaking (thermodynamic bifurcation; “chiral sign selection”). The low values of hydrodynamic forces do not justify a classical chiral induction. Therefore, the phrase in the Introduction (page 3 lines 18-19 and page 4 line one) can be only referred to ref. 34 but not to ref. 18. In contrast the lines 15-18 in page 3 should contain the citation 18 in the context of the effect of a chiral polarization in a spontaneous mirror symmetry breaking process. For example changing the text to: “For example, clockwise (CW) or counterclockwise (CCW) stirring of a solution containing exclusively protonated achiral porphyrin molecules leads to the formation of J-aggregates of opposite handedness. The phenomenon has been interpreted as the effect of the chiral polarization in a mirror symmetry breaking scenario^{18,23} but also that macroscopic hydrodynamic forces could shift the equilibrium of a racemic mixture of J-aggregates, driving a redistribution of chiral aggregates favored by the stirring^{18,(deleted),34}”.

Response: We thank the reviewer for this important suggestion. The following text is revised in the manuscript.

For example, clockwise (CW) or counterclockwise (CCW) stirring of a solution containing exclusively protonated achiral porphyrin molecules in cuvettes leads to the formation of J-aggregates (dye aggregates with an absorption band shifted to a longer wavelength of increasing sharpness than that of the monomer³¹) of opposite handedness after removing the vortex motion. The phenomenon has been interpreted as the effect of the chiral polarization in a mirror symmetry breaking scenario^{13,32}, but also that macroscopic (length scale larger than cm) hydrodynamic forces could shift the equilibrium of a racemic mixture of J-aggregates, driving a redistribution of chiral aggregates favored by the stirring³³.

Comment 1-2: Notice too, that some authors interpreted the CD signal arising by the macroscopic chiral ordering in a macroscopic vortex (refs. 17, 21), but that only

appears in dynamic stirring conditions. A final demonstration on this has been reported in *Nanoscale* 2015, 7, 20435. Obviously the objective of the paper is not a critical review of the topic, where the authors show important contributions, but it could be referred that the different interpretations of the phenomenon have been discussed in *The Chemical Record* 2017,17,1-13. This citation should substitute the older and less critical revision of ref. 23 (*Chem Soc Rev*).

Response: We thank the reviewer for this critical comment. The interpretation of CD signals in dynamic stirring conditions is added in the revised manuscript.

Moreover, the enantiospecific reversible effect on J-aggregates in dynamic stirring conditions is observed as a result of the chiral ordering of high aspect ratio nanoparticles in the flows, other than the appearance of intrinsically chirality^{34,35}.

Comment 1-3: In respect to the thermodynamic nature of the effect of the chiral polarization of stirring in a spontaneous mirror symmetry breaking process, this should be described in a more accurate way. For example in the caption of Fig. 1 the text (page 5, lines 7 and 8) “b, Microvortex-induced symmetry breaking of CW- or CCW-rotated achiral...” must be corrected, for example to “b, Effect of the shear forces of enantiomorphic microvortices (CW and CCW rotating directions) on the mirror symmetry breaking process of achiral BTAC molecules in DMF/H₂O mixtures leading to the formation of BTAC gels”.

Response: We thank the reviewer for pointing this out. The caption of Fig. 1 is revised.

b, Effect of the shear forces of enantiomorphic microvortices (CW and CCW rotating directions) on the mirror symmetry breaking process of achiral BTAC molecules in DMF/H₂O mixtures leading to the formation of BTAC gels.

Comment 1-4: Further, such unclear descriptions of the mirror symmetry breaking also appear in the Abstract. The abstract should be rewritten. Now it does not? show the same quality level than that of the reported work.

Response: We thank the reviewer for this important comment. The abstract is rewritten.

The origin of homochirality in life is a fundamental mystery. Symmetry breaking and subsequent amplification of chiral bias are regarded as one of the underlying

mechanisms. However, the selection and control of initial chiral bias in a spontaneous mirror symmetry breaking process remains a great challenge. Here we provide the first experimental evidence that laminar chiral microvortices generated within asymmetric microchambers can lead to a hydrodynamic selection of initial chiral bias of supramolecular systems composed of exclusively achiral molecules within milliseconds. The self-assembled nuclei with the chirality sign affected by the hydrodynamic shear force from microvortices are subsequently amplified into almost absolutely chirality-controlled supramolecular gels or nanotubes. In contrast, turbulent vortices in stirring cuvettes fail to select the chirality of supramolecular gels.

Comment 2-1: The viscous drag generates the vortices, but for the chiral effect on the self-assembly the cause “upon” the particle is that of the shear forces. Laminar flows are necessary for the alignment of the growing particles, and the particle shape and length must be able to detect the velocity gradients. Viscous drag is the way how the device creates the vortexes but not the cause of the chiral effect upon the aggregate particles, which is the particle alignment and the existence of a strong enough torque. This is not clear in some parts of the text and should be revised.

Response: We thank the review for pointing this out. To clarify, the effects of laminar flows on particle alignment and twist are added.

These instantly formed nuclei are aligned in the laminar flows, and be able to detect the velocity gradients due to the increase in size. A shear force (F_s) originated from the shear rate gradient is applied to the nuclei, with the maximum of 1.6 pN for BTAC nuclei and 0.1 pN for TPPS₄ nuclei on the basis of the maximal shear rate gradient of $8 \times 10^{10} \text{ m}^{-1} \text{ s}^{-1}$.

Comment 2-2: The fig. 5 only tries to show the shear gradient in the Z and X direction of the microflow chamber, but for a chiral effect it must also exist a gradient in the Y direction. We surely agree with the authors that the gradients originating the torque on the particle aligned in the flow statistically follow the vectorial description of

I don't think that the present text gives a clear insight on this for a reader not working

directly in the field.

Response: We thank the reviewer for this important suggestion. The spatial distribution of the shear rate gradient is shown in the revised Fig. 5.

Figure 5 | a, Schematic mechanism of laminar chiral microvortex-selected chirality of BTAC and TPPS₄ nuclei. The rapidly formed nuclei (< 1 ms) are aligned in the laminar flow, and twisted by a shear force (F_s) to give rise to an initial chiral bias dependent on the rotation sense of microvortices. These primary nuclei then serve as the templates for the subsequent growth into supramolecular assemblies with the predefined chirality. The spatial distribution of the shear rate gradient is determined by CFD simulations.

Comment 3: Some references of previous reports describing similar devices with lateral microchambers for the generation of vortices, should be cited, but pointing their differences in respect to chamber device reported here.

Response: We thank the reviewer for this insightful comment. Some previous works describing laminar microvortices for cell capture are cited.

Previous investigations show that laminar microvortices arising from the sudden expansion-contraction regions in microchannels could be used for size-based capture and concentration of rare cancer cells in blood samples^{47,48}. However, these microvortices are not chiral because of the symmetric shear rate gradients in the z

direction in vertical microchannels, and may not be capable of exerting enantioselective controls on supramolecular aggregates.

Comment 4: In respect which the structural changes that may occur at the bifurcation point by effect of the hydrodynamic torques, I would like to comment some points, independently if they are taken or not into account in the revised text. The primary structure of the sulfonatoporphyrin J aggregates is that of a sheet (J. Mater. Sci. C 2013, 1, 33377) where the order porphyrin building blocks is of conformational racemic compositions, but the porphyrins show two twists, one inclination in the plane and other out of the sheet plane of the sheet. This converts the supramolecular structure in intrinsically chiral. In my opinion the hydrodynamic effect should occur at this stage of the particle growth. These sheets then close to the hollow nanotubes or other type of particles (Chem Eur J 2016, 22, 9749). The first symmetry breaking transfers to other stereomorphic aspects of the particles (for example nanotubes). The work makes already some comments on the fact of such hierarchical chirality transfer. I suppose that the authors could emphasize this in some parts of the text.

Response: We thank the reviewer for this critical comment. The hydrodynamic effect on the hierarchical chirality of hollow nanotubes enclosed by the porphyrin sheets is discussed.

We should note that the TPPS₄ monomers may also self-assemble into the two-dimensional flexible sheets with intrinsically chirality, and roll into the short nuclei of monolayered nanotube structures by the hydrodynamic effect^{45,46}.

Comment 5: Less important: The groups of accumulated cites (for example in page 3 13-15, 16-23, 24-28, etc) should appear in the References section in chronological order.

Response: We thank the reviewer for this suggestion. All the accumulated cites are organized in chronological order.

Response to Reviewer #2

Comment: The possible origins of prebiotic homochirality is a topic of great current interest. Stirring of solutions of achiral porphyrins has previously been shown to induce chiral supramolecular systems of J-aggregates with opposite handedness for clockwise or counterclockwise stirring. Although stirring of gels also generates supramolecular chirality, the handedness is determined stochastically rather than by the sense of stirring. The present paper extends this work to microvortices generated in asymmetric microchambers via a clever microfluidic device that produces supramolecular aggregates which, remarkably, show right- or left-handed chirality depending on the rotation sense for both solutions and gels. The experimental SEM data showing M and P twists of resulting BTAC gels are convincing, as are the associated mirror-image CD spectra which are reproducible over 27 out of 28 experiments. Likewise for the CD spectra of the generated J-aggregates, being reproducible over 28 out of 28 experiments. However, I do not have the expertise to assess the computational fluid dynamics simulations. A plausible scenario is suggested in which natural rock micropores might support this novel far-from-equilibrium enantioselection process, thereby providing possible new insight into the natural origin of homochirality. This well-written paper should be of interest to a wide audience.

Response: We thank the reviewer for examining our research work, and his/her positive comment. A plausible scenario in which natural rock micropores might support this far-from-equilibrium enantioselection process is provided upon the reviewer's request.

The natural origin of homochirality is a long-standing question. Despite most of the biological chirality observed on Earth being one-signed, both left- and right-handed polyproline II helices are commonly found in globular proteins⁴⁹. It has been proposed that symmetry breaking at the molecular level induced by an external chiral influence may lead to a small excess of one enantiomer, which can be significantly amplified by autocatalytic reactions to display homochirality under prebiotic environments^{50,51}. Our study envisions a plausible scenario that prebiotic molecules could self-assemble into small nuclei/aggregates when passing through the natural asymmetric rock micropores in which chiral laminar microvortices exert a hydrodynamic torque on the nuclei. This chiral bias of nuclei appears almost instantly while being correlated with the rotation sense of microvortices. The subsequent propagation of this initial enantiomeric imbalance could result in the emergence of homochirality. This is analogous to a hypothesis that enantiomeric excesses might be induced by circularly polarized vacuum ultraviolet from space⁵².

Response to Reviewer #3

Comment: This paper offers interesting insights and experiments demonstrating the controlled formation of chiral aggregates in flows in microfluidic devices there the sense of the chirality is set by the sign of the vorticity in the microchambers. Results are shown for two different molecules, as given in Figure 2 for one molecule and Figure 3 for a second molecule. I do not know the background of this topic to know whether all of the results are original nor do I know whether it is significant that the results shown here are with gels whereas apparently some of flow drive chirality in solutions (in contrast to gels) seems to have been demonstrated (see item I below). The work is thought-provoking and I can see it being of interest to readers. There are four ideas that need clarification:

Response: We thank the reviewer for the positive evaluation of our research work. The following comments have been clarified upon the reviewer's request.

Comment 1: I sensed that the ideas likely are new but then I am a little confused by some of the references and descriptions, such as are not the experiments shown in this paper similar to the references made in the statement (p. 3) "For example, clockwise (CW) or counterclockwise (CCW) stirring of a solution containing exclusively protonated achiral porphyrin molecules leads to the formation of J-aggregates of opposite handedness." Hence, isn't this essentially what the authors say they accomplish when they write (p. 4): "we designed a microfluidic device to induce microscopic chiral factors, counter-rotating chiral microvortices (CW and CCW) inside opposing microchambers, to control over the supramolecular chirality of assemblies either in gels or solutions, exclusively from achiral molecules (Fig. 1a and Supplementary Fig. 1)." So, I think it needs to be more clear whether and how the flow-driven chirality, so nicely demonstrated in this paper, is new?

Response: We thank the reviewer for this important comment. As commented by Reviewer 1, the relatively few previous results obtained by shear hydrodynamic forces have been object of quite different explanations, even of skeptical considerations that have hindered the progress in the field. How such chiral hydrodynamic forces could be created was quite unclear and the submitted work gives an answer to the problem; and the answer opens a broad prospect of future innovations. In respect to the today available methods the work represents a qualitative change in the experimental stereochemical control of the formation of supramolecular dissipative structures (open thermodynamic systems). To clarify, the following text has been revised in the manuscript.

For example, clockwise (CW) or counterclockwise (CCW) stirring of a solution containing exclusively protonated achiral porphyrin molecules in cuvettes leads to the formation of J-aggregates (dye aggregates with an absorption band shifted to a longer wavelength of increasing sharpness than that of the monomer³¹) of opposite handedness after removing the vortex motion. The phenomenon has been interpreted as the effect of the chiral polarization in a mirror symmetry breaking scenario^{13,32}, but also that macroscopic (length scale larger than cm) hydrodynamic forces could shift the equilibrium of a racemic mixture of J-aggregates, driving a redistribution of chiral aggregates favored by the stirring³³. Moreover, the enantiospecific reversible effect on J-aggregates in dynamic stirring conditions is observed as a result of the chiral ordering of high aspect ratio nanoparticles in the flows, other than the appearance of intrinsically chirality^{34,35}. In contrast to the flow-controlled chirality in solutions, stirring during gelation fails to affect the supramolecular chirality in gels, which has been determined stochastically by initial spontaneous symmetry breaking⁶. On the basis of these contradictory observations, it remains an enigma how an efficient hydrodynamic force could be created and how it controls over the supramolecular chirality during the self-assembly process.

Here we provide the first experimental evidence that laminar chiral microvortices generated within asymmetric microchambers can lead to a hydrodynamic selection of initial chiral bias of supramolecular systems composed of exclusively achiral molecules within milliseconds.

In contrast, turbulent vortices in stirring cuvettes fail to select the chirality of supramolecular gels.

Comment 2: Second, in the conclusions the authors write "We anticipate that our investigation may provide a hint as to the origins of natural homochirality, i.e. the critical role of oceanic vortices in protein folding and self-assembly in early Earth." But isn't the biological chirality observed on Earth largely one-signed? How do the authors think a flow feature, which would produce both left- and right-handed flows (signed vortices) could be responsible for selecting sign? Much as I find the demonstration of flow-driven chirality interesting, I think that it is unfair to claim relevance to biology without some plausible remarks about selection of "sign".

Response: We thank the reviewer for this critical comment. The origin of homochirality in biological systems is a long-standing question. Despite most of the biological chirality observed on Earth being one-signed, both left- and right-handed

polyproline II helices are commonly found in globular proteins⁴⁹. It has been proposed that the homochirality may have begun with mirror symmetry breaking and a subsequent amplification of the chiral bias. In other words, a small excess of one enantiomer caused by an external chiral influence could be significantly amplified by autocatalytic reactions to display homochirality under prebiotic environments. To clarify, the follow text is added into the revised manuscript.

The natural origin of homochirality is a long-standing question. Despite most of the biological chirality observed on Earth being one-signed, both left- and right-handed polyproline II helices are commonly found in globular proteins⁴⁹. It has been proposed that symmetry breaking at the molecular level induced by an external chiral influence may lead to a small excess of one enantiomer, which can be significantly amplified by autocatalytic reactions to display homochirality under prebiotic environments^{50,51}. Our study envisions a plausible scenario that prebiotic molecules could self-assemble into small nuclei/aggregates when passing through the natural asymmetric rock micropores in which chiral laminar microvortices exert a hydrodynamic torque on the nuclei. This chiral bias of nuclei appears almost instantly while being correlated with the rotation sense of microvortices. The subsequent propagation of this initial enantiomeric imbalance could result in the emergence of homochirality. This is analogous to a hypothesis that enantiomeric excesses might be induced by circularly polarized vacuum ultraviolet from space⁵².

Comment 3: Third, maybe I misunderstood the SI but in supplementary figure 4 we read that a chiral-like signal is only obtained at 40 °C while in supplementary figure 5 a chiral-like signal is only obtained at one concentration. If so, this point is not clearly made in the main text. It also highlights that the "critical conditions" remain to be identified, which I can appreciate is a much longer term project but is something the authors can indicate to the reader.

Response: We thank the reviewer for pointing this out. The critical conditions in the microfluidic settings are highlighted in the revised manuscript.

The optimization experiments are carried out to identify the critical conditions (BTAC concentration of 61 mg/mL and temperature of 40 °C) in which the chirality of supramolecular systems is dependent on the rotation direction of microvortices (Supplementary Figs. 4 and 5).

Comment 4: In my view the authors misuse the terms "microscopic" and "macroscopic" when discussing the features of the flow and their experiments. In

fluid mechanics, as I think the authors realize given some of their discussion, a flow is characterized by the Reynolds number. Even "microscopic" systems, by which the authors may refer to systems that have dimensions 100s of microns can be turbulent if the flow is fast enough just as large-scale "macroscopic" flows can be laminar if the flow is slow enough (e.g. flows of glaciers). This microscopic/macroscopic language is often misused in the literature but here it is particularly unwise since (as supported by figure 5 in contrast to figures 2 and 3) the distinction in the authors results appears to be between laminar and turbulent flows (rather than between microscopic and macroscopic). It is also for this reason that the authors have to clarify item I above to demonstrate that their work is really new and not simply a "small scale" version of flow-driven chirality that already may have been demonstrated.

I believe this nice paper has a potentially original message to share but I think points I-IV above need clarification so that the results, previous work, and potential claims to the origin of chirality in biological systems are put into proper perspective.

Response: We thank the reviewer for this important comment. We agree with the reviewer that we misuse the terms "microscopic" and "macroscopic" when discussing the features of the flow. To clarify, we revise the subtitle in Fig. 5 and the related descriptions in the text.

Fig. 5a. Laminar microvortices in microchambers; Turbulent vortices in stirring cuvettes.

Comment 5: p. 3: What are "J-aggregates"?

Response: We thank the reviewer for this question. The explanation of J-aggregates is added into the text.

J-aggregates (dye aggregates with an absorption band shifted to a longer wavelength of increasing sharpness than that of the monomer³¹)

Comment 6: p. 4: "Microscopic chiral influences, such as chiral microflow, may impact on the selection of supramolecular chirality on a similar scale, and yet evidence is relatively scarce." - what does this mean? Has evidence been published? Since "micro" is not defined, this sounds similar to me to the previous statements in the introduction (p. 3, though "macro" is not defined): "Indeed, purely physical fields, such as hydrodynamic flow by stirring¹⁶⁻²³, circularly polarized light irradiation²⁴⁻²⁸, or the combination of hydrodynamic, magnetic, thermal, or light factors^{9,29-33} can cause enantioselection in racemic mixtures by inducing an

enantiomeric excess."

Response: We thank the reviewer for pointing this out. To clarify, we delete the statement regarding the microscopic chiral influences in the Introduction. The definitions of microvortices and macroscopic are added in the text.

microvortices (length scale from tens to hundreds of μm)
macroscopic (length scale larger than cm)

Comment 7: In fact, the remark about flows producing chiral structures from achiral starting features appears identical to me to one demonstration in this paper (p. 4): "The chiral microvortices enabled a primary selection of the chiral bias dependent on rotation sense".

Response: We thank the reviewer for this comment. The statement regarding the chiral microvortices is revised.

The high magnitude of the shear gradient in laminar microvortices allows the alignment and twist of the supramolecular aggregates against the Brownian regime when mirror symmetry breaking occurs.

Comment 8: p. 4: What are "M" and "P" chirality? This should be defined in the text (instead of the caption to figure 1).

Response: We thank the reviewer for this suggestion. The *M/P* terminology is used particularly for molecules that actually resemble a helix with a left/right handed chirality. The definition of *M/P* is added in the text.

left-handed helical twists (*M* chirality)
right-handed helical twists (*P* chirality)

Comment 9: "which have been reported to exhibit symmetry breaking during gel formation or aggregation" - what are the references? I guess they mean reference 35, which is indicated earlier in the sentence, but this is rather disingenuous for an article that uses a microfluidic flow to show (the title of the 2016 article) "Milliseconds make the difference in the far from equilibrium assembly of supramolecular chiral nanostructures".

Response: We thank the reviewer for this comment. Two references about the

symmetry breaking of BTAC and TPPS₄ are cited.

Two kinds of achiral molecules, (tris(ethyl cinnamate) benzene-1,3,5-tricarboxamide, BTAC⁶, and tetra-(4-sulfonatophenyl) porphyrin), TPPS₄¹³, which have been reported to exhibit symmetry breaking during gel formation or aggregation, are selected.

Comment 10: P. 6: what is a "negative Cotton effect"?

Response: We thank the reviewer for this question. The Cotton effect is the characteristic change in optical rotatory dispersion and/or circular dichroism in the vicinity of an absorption band. Cotton effect is termed as negative when the negative peak first appears as the wavelength decreases.

The circular dichroism (CD) spectra show a negative Cotton effect (the negative peak first appears as the wavelength decreases) for the L-outlet and a positive Cotton effect (the positive peak first appears as the wavelength decreases) for the R-outlet (Fig. 2b), while no linear dichroism (LD) effects are detected in BTAC gels (Supplementary Fig. 6).

Comment 11: p. 6: "Predominant amounts" - a quantitative approximation should be indicated.

Response: We thank the reviewer for this important comment. A quantitative approximation of left- or right-handed twists in BTAC gels is determined using the SEM images.

Scanning electron microscopy (SEM) analysis indicates that the percentage of left-handed helical twists (*M* chirality) is 80 % (n = 253) in BTAC gels from the L-outlet, and that of right-handed helical twists (*P* chirality) is 86 % (n = 271) in gels from the R-outlet (Figs. 2c-d and Supplementary Fig. 7).

Supplementary Figure 7 | SEM images of BTAC gels. Predominant amounts of (a) left- (*M*, 80 %) and (b) right-handed (*P*, 86 %) twists are observed in BTAC gels from the L- or R-outlet.

Comment 12: Figure 2 is a very nice demonstration that the local flow controls the local chirality of the aggregates.

Response: Thanks for the comment.

Comment 13: p. 12-13: The microfluidic chamber width is given but what is the height? If possible, give the dimensions in figure 4. As far as I can tell from the SI, the height of the channels was 50 microns.

Response: We thank the reviewer for this comment. The height of the microchamber is added in the revised Fig. 4a.

Figure 4 | Analysis of laminar chiral microvortices. a, CFD simulation shows that fluid can rotate upward (red) or downward (blue) to generate laminar *P*- or *M*-chiral microvortices in the L-microchamber. The channel height is 50 μm.

Comment 14: p. 13: The statement about turbulence with a Reynolds number of 2000 applies to flow in a straight pipe and is not the right number of turbulence in flows with curved streamlines, which likely transitions to turbulence at a lower Reynolds number.

Response: We thank the reviewer for this important comment. The Reynolds number of the microchamber is calculated as 380, which is lower than the Reynolds number of ~ 470 of the main straight microchannel. In addition, the flow visualization in the microchannels shows a clear boundary between the microvortices and the ambient fluid, indicating that the flow is laminar within the microchambers (Figure R1). To clarify, the statement about turbulence is revised.

Figure R1 Visualization of fluid motion using xylenol orange dye (50 mM). Flow condition: 1 mL/h for the middle inlet and 30 mL/h for each side inlet.

The Reynolds number of the main channel was calculated as

$$\text{Re} = \frac{U_{\max} D_h}{\nu} = \frac{7\text{ms}^{-1} \times 66.7\mu\text{m}}{10^{-6}\text{m}^2\text{s}^{-1}} = 467$$

where U_{\max} is the maximum channel velocity, D_h is the hydraulic diameter of the main channel that is expressed as $D_h = 2WH/(W + H) = 66.7\mu\text{m}$, and ν is the kinematic viscosity.

The Reynolds number of the microchamber was calculated as

$$\text{Re} = \frac{U_{\text{cham}} W_{\text{cham}}}{\nu} = \frac{1.9\text{ms}^{-1} \times 200\mu\text{m}}{10^{-6}\text{m}^2\text{s}^{-1}} = 380$$

where U_{cham} is the average flow velocity at the junction of the microchamber and the main channel and W_{cham} is chamber width.

Despite the high rotation speed, these microvortices remain laminar with the Reynolds number of ~ 470 of the main straight channel and that of 380 of the microchamber (Supplementary Information, and Supplementary Fig. 3).

Comment 15: The shear rate gradient in figure 4c?

Response: We thank the reviewer for this comment. The determination of the shear rate gradient is presented in the Supporting Information.

The shear rate gradients in the x , y , and z directions were calculated as $\partial\gamma_x/\partial x$, $\partial\gamma_y/\partial y$, and $\partial\gamma_z/\partial z$, respectively. Here γ_x , γ_y , and γ_z were the shear rates accounting for the fluid rotation about the x , y , and z axes and expressed as $\partial u_z/\partial y - \partial u_y/\partial z$, $\partial u_x/\partial z - \partial u_z/\partial x$, and $\partial u_y/\partial x - \partial u_x/\partial y$, respectively.

Comment 16: p. 12: what does "fluid rotations travelled upward" mean?

Response: We thank the reviewer for pointing this out. To clarify, this description is revised.

Interestingly, the majority ($\sim 84\%$) of microvortices in the L- (R-) microchamber tend

to spiral upward, showing a predominantly *P* (*M*) chirality (Fig. 4b).

Comment 17: "new paradigm of chiral selection of supramolecular assemblies at the initial stage of self-assembly." - I agree this is shown but it is consistent

Response: We thank the reviewer for the comment. This statement is revised.

These rapidly-rotating laminar chiral microvortices are of significance in the selection of chirality at the initial stage of self-assembly.

Comment 18: The assembly time is estimated as 0.7 ms for BTAC and 0.5 ms for TPPS₄ - how is this estimated? How can the simulation of flow say anything about "assembly"? In the SI (p. 6) we read that the assembly time is calculated as $L^2/(6D)$ where *L* is the size of the primary nuclei but, while I agree this is a diffusion time, why should this necessarily be the assembly time?

Response: We thank the reviewer for the comment. The self-assembly process is initiated spontaneously when the achiral BTAC (TPPS₄) building blocks are rotated within the chiral microvortices, and diffuse into the DMF/H₂O (HCl/C₂mim⁺) solutions. Therefore, we estimate the assembly time using the equation of the diffusion time. To clarify, an explanation is added in the Supporting Information.

The self-assembly process was initiated spontaneously when the achiral BTAC (TPPS₄) building blocks were rotated within the chiral microvortices, and diffused into the DMF/H₂O (HCl/C₂mim⁺) solutions. Therefore, we estimated the assembly time using the equation of the diffusion time $t = L^2/6D$.

Comment 19: The authors write that the aggregates experience a "drag force" - this is not true in general for neutrally buoyant objects in flow. In the absence of a body force such as gravity these objects are force free. In the SI gravity is indicated but isn't this force much smaller than the other forces included in the simulation? Perhaps the point is that there are intermolecular forces with other molecules and then there can be a balancing fluid drag force. So the text has to be clarified.

Response: We thank the reviewer for this important comment. We agree with the reviewer that there is no drag force exerting on the aggregates. We thus use the term "shear force" in the revised text.

Comment 20: Also, in the SI a drag force is calculated as $A\eta L\Gamma/2$ where Γ is the shear rate gradient - why is this quantity a drag force? I agree that the quantity has units of force but what fluid mechanical result indicates that this quantity is a drag force and is there a reference for this result or is it just an order of magnitude idea?

Response: We thank the reviewer for this insightful comment. The clarification is added in the Supporting Information.

The hydrodynamic torque to twist the nuclei was originated from the shear force. For a TPPS₄/BTAC nucleus with a length of L suspending in a flow field with a shear rate gradient Γ , the shear force exerting on the nucleus was estimated as

$$F_d = A\eta L\Gamma/2.$$

Comment 21: The twist on the bottom of nuclei was stronger than that on the crown, attributed to the decreased F_d along the increased microchamber height, resulting in the opposite signs between the chiral bias of nuclei and the rotation direction of microvortices (Fig. 5a).

Response: We thank the reviewer for the comment. We revise F_d (drag force) to F_s (shear force) in the Fig. 5a and the text.

The twist on the bottom of nuclei is stronger than that on the crown, attributed to the decreased F_s along the increased microchamber height, resulting in the opposite signs between the chiral bias of nuclei and the rotation direction of microvortices (Fig. 5a).

Comment 22: In the concluding paragraph the authors return to remark about a possible role of rock micropores and refer to their experiments with 10^4 rpm. Is this realistic for flows over rock micropores?

Response: We thank the reviewer for this insightful comment. It is realistic for flows over rock micropores at a high rotation speed up to 10^4 rpm. Upon the reviewer's request, the illustration and the relevant references are added in the text.

On the prebiotic Earth, rock micropores in the size range from tens to hundreds of μm are ubiquitous in the vicinity of hydrothermal vents in the ocean³⁶. The velocity of flow near the vents could reach several meters per second, and there is a large change to create high speed microvortices (length scale from tens to hundreds of μm) up to

10⁴ rpm for flows over rock micropores³⁷.

Comment 23: 30 mL/hr; according to p. S1 of the SI the height of the channels was 50 microns and from figure 5 the widths of the microchamber and the main channel appear to be about 100 microns. The typical maximum speed in the main channel is then about

$$\frac{30\text{cm}^3}{3600\text{s} \times (50 \times 100)\mu\text{m}^2} = 3\text{ms}^{-1}$$

Based on this estimate, the experiments would have a typical

$$\text{Re} = \frac{2.5\text{mm} \times 3\text{ms}^{-1}}{10^{-6}\text{m}^2\text{s}^{-1}} = 7500$$

which is probably somewhat of an overestimate.

The maximum gradient expected is then $3\text{ms}^{-1}/25\mu\text{m} = 10^5\text{s}^{-1}$. These estimates seem comparable to the numerical results as far as I can tell.

Response: We thank the reviewer for this comment. We have carefully examined the Reynolds number in the microchannel. The average speed in the main channel is

$$U_{\text{aver}} = \frac{Q_{\text{total}}}{WH} = \frac{61\text{mL/h}}{(100\mu\text{m} \times 50\mu\text{m})} = \frac{61 \times 10^{-6}\text{m}^3 \div 3600\text{s}}{(100 \times 50) \times 10^{-12}\text{m}^2} = 3.3\text{ms}^{-1}$$

where the total flow rate is 61 mL/h, resulting from 30 mL/h for each of the two side inlets and 1 mL/h for the middle inlet.

Using numerical simulation, we determine the maximum velocity as $\sim 7\text{ms}^{-1}$. We use hydraulic diameter D_h as the characteristic dimension of the microchannel

$$D_h = \frac{2WH}{(W + H)} = 66.7\mu\text{m}$$

We subsequently calculate the Reynolds number as

$$\text{Re} = \frac{U_{\text{max}} D_h}{\nu} = \frac{7\text{ms}^{-1} \times 66.7\mu\text{m}}{10^{-6}\text{m}^2\text{s}^{-1}} = 467$$

which is consistent with our claim in the manuscript.

We have added the calculation of the Reynolds number in the revised Supplementary Information.

The Reynolds number of the main channel was calculated as

$$\text{Re} = \frac{U_{\max} D_h}{\nu} = \frac{7 \text{ms}^{-1} \times 66.7 \mu\text{m}}{10^{-6} \text{m}^2 \text{s}^{-1}} = 467$$

where U_{\max} is the maximum channel velocity, D_h is the hydraulic diameter of the main channel that is expressed as $D_h = 2WH/(W + H) = 66.7 \mu\text{m}$, and ν is the kinematic viscosity.

Comment 24: The flow simulations seem fine. The specific values given show the idea and indicate that future work will be needed to properly characterize critical conditions for the flow-driven assembly in terms of appropriate dimensionless parameters.

Response: We thank the reviewer for this insightful comment. The discussion is added in the text.

These simulation and experimental results show that future work will be needed to properly characterize critical conditions for the flow-driven assembly in terms of appropriate dimensionless parameters.

References

- 6 Shen, Z., Wang, T. & Liu, M. Macroscopic Chirality of Supramolecular Gels Formed from Achiral Tris (ethyl cinnamate) Benzene - 1, 3, 5 - tricarboxamides. *Angew. Chem.-Int. Edit.* **53**, 13424-13428 (2014).
- 13 Ribó, J. M., Crusats, J., Sagués, F., Claret, J. & Rubires, R. Chiral sign induction by vortices during the formation of mesophases in stirred solutions. *Science* **292**, 2063-2066 (2001).
- 31 Würthner, F., Kaiser, T. E. & Saha - Möller, C. R. J - Aggregates: From Serendipitous Discovery to Supramolecular Engineering of Functional Dye Materials. *Angew. Chem.-Int. Edit.* **50**, 3376-3410 (2011).
- 32 Crusats, J., El-Hachemi, Z. & Ribó, J. M. Hydrodynamic effects on chiral induction. *Chem. Soc. Rev.* **39**, 569-577 (2010).
- 33 D'Urso, A., Randazzo, R., Lo Faro, L. & Purrello, R. Vortexes and nanoscale chirality. *Angew. Chem.-Int. Edit.* **49**, 108-112 (2010).
- 34 Arteaga, O., Canillas, A., El-Hachemi, Z., Crusats, J. & Ribo, J. M. Structure vs. excitonic transitions in self-assembled porphyrin nanotubes and their effect on light absorption and scattering. *Nanoscale* **7**, 20435-20441 (2015).
- 35 Ribo, J. M., El - Hachemi, Z., Arteaga, O., Canillas, A. & Crusats, J. Hydrodynamic Effects in Soft - matter Self - assembly: The Case of J - Aggregates of Amphiphilic Porphyrins. *Chem. Rec.* **17**, 713-724 (2017).

- 36 Kelley, D. S. *et al.* An off-axis hydrothermal vent field near the Mid-Atlantic Ridge at 30° N. *Nature* **412**, 145 (2001).
- 37 Ginster, U., Mottl, M. J. & Herzen, R. P. V. Heat flux from black smokers on the Endeavour and Cleft segments, Juan de Fuca Ridge. *J. Geophys. Res.-Solid Earth* **99**, 4937-4950 (1994).
- 42 Sorrenti, A., Rodriguez-Trujillo, R., Amabilino, D. B. & Puigmartí-Luis, J. Milliseconds Make the Difference in the Far-from-Equilibrium Self-Assembly of Supramolecular Chiral Nanostructures. *J. Am. Chem. Soc.* **138**, 6920-6923 (2016).
- 45 El-Hachemi, Z. *et al.* Structure vs. properties - chirality, optics and shapes - in amphiphilic porphyrin J-aggregates. *J. Mater. Chem. C* **1**, 3337-3346 (2013).
- 46 El - Hachemi, Z. *et al.* Effect of Hydrodynamic Forces on meso - (4 - Sulfonatophenyl) - Substituted Porphyrin J - Aggregate Nanoparticles: Elasticity, Plasticity and Breaking. *Chem.-Eur. J.* **22**, 9740-9749 (2016).
- 47 Sollier, E. *et al.* Size-selective collection of circulating tumor cells using Vortex technology. *Lab Chip* **14**, 63-77 (2014).
- 48 Khojah, R., Stoutamore, R. & Di Carlo, D. Size-tunable microvortex capture of rare cells. *Lab Chip* **17**, 2542-2549 (2017).
- 49 Adzhubei, A. A. & Sternberg, M. J. E. Left-handed Polyproline II Helices Commonly Occur in Globular Proteins. *J. Mol. Biol.* **229**, 472-493 (1993).
- 50 Plasson, R., Kondepudi, D. K., Bersini, H., Commeyras, A. & Asakura, K. Emergence of homochirality in far - from - equilibrium systems: Mechanisms and role in prebiotic chemistry. *Chirality* **19**, 589-600 (2007).
- 51 Kafri, R., Markovitch, O. & Lancet, D. Spontaneous chiral symmetry breaking in early molecular networks. *Biol. Direct* **5**, 38 (2010).
- 52 Meierhenrich, U. J. *et al.* Asymmetric Vacuum UV photolysis of the Amino Acid Leucine in the Solid State. *Angew. Chem.-Int. Edit.* **44**, 5630-5634 (2005).

REVIEWERS' COMMENTS:

Reviewer #1 (Remarks to the Author):

The authors have improved the manuscript in the present revised form. My previous comments have been fully addressed.

I encourage the editorial office for the acceptance of this singular and innovative work.

Reviewer #2 (Remarks to the Author):

This revised paper generally looks fine to me, although my lack of relevant expertise prevents me from commenting on most of the responses. However, I can say that one new addition in the revised manuscript is incorrect. Thus in response to comment 2 of reviewer 3, the authors have added

“...both left- and right-handed polyproline II helices are commonly found in globular proteins (ref. 49)”.

In fact, as detailed in ref. 49, polyproline II helices occur exclusively as left-handed. The authors may be confused with alpha-helices, which have a distinct conformation that is predominantly right-handed, although a rare left-handed version does sometimes occur in proteins; whereas polyproline II helices are exclusively left-handed. All protein secondary structure elements originate in polypeptide chains made up entirely of L-amino acids. So this new passage, together with ref. 49, should be deleted.

Reviewer #3 (Remarks to the Author):

The authors have made a significant effort in the revised paper to clarify the work and I think they have done a good job. I read the paper again and noted a few sentences that seemed unclear to me. Otherwise I recommend publication.

Remarks:

1) p. 3: "but also that macroscopic (length scale larger than cm)" - I am skeptical of how a distinct length scale can be assigned but admittedly I am not familiar with the details of the experimental paper that is cited.

2) p. 4: I can't make sense of the final phrase in "Moreover, the enantiospecific reversible effect on J-aggregates in dynamic stirring conditions is observed as a result of the chiral ordering of high aspect ratio nanoparticles in the flows, other than the appearance of intrinsically chirality^{34,35}".

3) p. 10: "Notice that the vortices in stirring cuvettes at ~1000 rpm are of highly turbulence with the Reynolds number". I suppose the authors mean "... are turbulent with the Reynolds ..."

Response to Reviewer #1

Comment: The authors have improved the manuscript in the present revised form. My previous comments have been fully addressed.

I encourage the editorial office for the acceptance of this singular and innovative work.

Response: We thank the reviewer for the positive evaluation of our research work, and the suggestion of acceptance of this work.

Response to Reviewer #2

Comment: This revised paper generally looks fine to me, although my lack of relevant expertise prevents me from commenting on most of the responses. However, I can say that one new addition in the revised manuscript is incorrect. Thus in response to comment 2 of reviewer 3, the authors have added “...both left- and right-handed polyproline II helices are commonly found in globular proteins (ref. 49)”. In fact, as detailed in ref. 49, polyproline II helices occur exclusively as left-handed. The authors may be confused with alpha-helices, which have a distinct conformation that is predominantly right-handed, although a rare left-handed version does sometimes occur in proteins; whereas polyproline II helices are exclusively left-handed. All protein secondary structure elements originate in polypeptide chains made up entirely of L-amino acids. So this new passage, together with ref. 49, should be deleted.

Response: We thank the reviewer for pointing out this issue. The sentence “Despite most of the biological chirality observed on Earth being one-signed, both left- and right-handed polyproline II helices are commonly found in globular proteins⁴⁹” has been deleted in the main text.

Response to Reviewer #3

Comment: The authors have made a significant effort in the revised paper to clarify the work and I think they have done a good job. I read the paper again and noted a few sentences that seemed unclear to me. Otherwise I recommend publication.

Response: We thank the reviewer for the positive evaluation of the revised

manuscript. The following comments have been clarified upon the reviewer's request.

Comment 1: p. 3: "but also that macroscopic (length scale larger than cm)" - I am skeptical of how a distinct length scale can be assigned but admittedly I am not familiar with the details of the experimental paper that is cited.

Response: We thank the reviewer for this important comment. To clarify, "length scale larger than cm" has been removed in the main text.

Comment 2: p. 4: I can't make sense of the final phrase in "Moreover, the enantiospecific reversible effect on J-aggregates in dynamic stirring conditions is observed as a result of the, other than the appearance of intrinsically chirality^{34,35}".

Response: We thank the reviewer for pointing this out. The sentence has been revised to "Moreover, the enantiospecific effect on J-aggregates in dynamic stirring conditions is reversible, due to the temporary chiral ordering of nanoparticles in the flows^{34,35}".

Comment 3: p. 10: "Notice that the vortices in stirring cuvettes at ~1000 rpm are of highly turbulence with the Reynolds number". I suppose the authors mean "... are turbulent with the Reynolds ...".

Response: We thank the reviewer for pointing this out. The sentence has been revised to "Notice that the vortices in stirring cuvettes at ~1000 rpm are turbulent with the Reynolds number...".